# On the Temporal Stability of People’s Annoyance with Road Traffic Noise

**DOI:** 10.3390/ijerph17041374

**Published:** 2020-02-20

**Authors:** Truls Gjestland

**Affiliations:** SINTEF DIGITAL, N-7465 Trondheim, Norway; truls.gjestland@sintef.no

**Keywords:** road traffic noise, annoyance, temporal trends

## Abstract

Sixty-one social surveys on annoyance caused by road traffic noise conducted world-wide over a period of forty-five years have been re-analyzed by various means for possible temporal trends. Eighteen of these surveys were conducted after 2000. People’s reactions to road traffic noise seem to have been stable across the study period. No indications were found that would warrant revision of the current EU reference curve for predicting the annoyance from road traffic noise.

## 1. Introduction

Some researchers assert that people today are more annoyed by road traffic noise than they were, say, 20–40 years ago [1,2] *inter alia*. Various subsets of recent and earlier annoyance surveys and various data analysis techniques have been used to justify such an assertion.

The annoyance caused by transportation noise is generally viewed as proportional to its long-term, cumulative noise level, but the prevalence of a consequential degree of traffic noise-induced annoyance varies considerably from one survey site to another.

Annoyance is expressed either on a linear scale extending from no annoyance at all to the most extreme annoyance, or, as is common practice for regulatory purposes, on a scale that shows the percentage of the exposed population that consider itself “highly annoyed”, HA. Highly annoyed refers to the upper 28% of the annoyance scale. The results from a survey on noise annoyance are presented as the percentage of highly annoyed people at a given noise exposure level. This treatment of noise exposure and annoyance values can be traced back to Schultz [3] and FICON—US Federal Interagency Committee on Noise [4] which declared that “Annoyance is its preferred summary measure of the general adverse reaction of people to noise, and that the percentage of the area population characterized as “highly annoyed” by long-term exposure to noise is its preferred measure of annoyance”.

For a typical noise exposure level, *L*_dn_ = 60 dB, the prevalence of highly annoyed residents can vary from 0% to 60% or more. Conversely a prevalence rate of 10% highly annoyed has been associated in various studies with noise levels ranging from about *L*_dn_ = 45 dB to *L*_dn_ = 75 dB. The analysis of temporal trends in the annoyance response is therefore highly dependent on the sub-set of survey data analyzed.

This paper analyzes a large set of reported surveys on annoyance from road traffic noise and compares results of various analysis techniques that have been reported in the scientific literature.

## 2. Materials and Methods

### Database

A database was compiled from surveys conducted during the period 1969–2014. References to older surveys were mainly found in [5]. Information on more recent surveys (after 2008) were collected via direct search of journal articles and conference proceedings. Only survey reports that provide information about noise exposure levels and corresponding prevalence of highly annoyed residents were considered.

Survey findings selected for the current analyses met the following criteria:Survey design similar to recommendations by International Commission on Biological Effects of Noise—ICBEN [6,7] or International Standards Organization—ISO [8].Noise measurements that could be converted to DNL or DENL.Definition of “highly annoyed” in compliance with a 70–75% cut-off of the annoyance scale.Findings reported as pairs of noise exposure and prevalence of annoyance.

No distinction was made between the two-common metrics for cumulative transportation noise exposure, DNL and DENL, as Brink et al. [9] have found that the difference between them is less than 0.5 dB for road traffic noise.

Table 1 shows the 61 surveys that were included in the current analysis, 18 of which were conducted in the last two decades.

The table lists the different studies by year of conduct. The “code” column refers to survey codes in [5]. This catalogue assigns a unique ID to most noise surveys conducted before 2008 for easy identification. The “EU” column indicates which surveys were included in the Miedema and Vos [52] analysis for the exposure–response curve currently in use by the EU. In addition to total numbers of respondents, the table also lists calculated community tolerance level—CTL value, and r^2^ for fitting the CTL function and a second order polynomial regression function to the original data set.

The total data set comprises about 600 paired observations of noise exposure and prevalence of high annoyance, based on the opinions of about 95,000 social survey respondents. More detailed information about the individual surveys can be found in the given references (see special list at the end).

## 3. Analysis Methods

### 3.1. Statistical Regression

The results of a survey on noise annoyance are typically presented as exposure–response relationships, also commonly referred to as dose–response or dosage–response functions. The results are plotted with the independent variable, noise exposure, measured as DNL, DENL, or a similar quantity, on the horizontal axis, and the prevalence of annoyance on the vertical axis.

Researchers who rely solely on regression analysis to support their conclusions typically enter all the field observations data into a standard statistical analysis software (e.g., SPSS—Statistical Package for the Social Sciences), and accept the output as a definite dosage–response without attempting to provide a logical explanation of the results and without regards for the implications of their analyses.

The results of analysis of individual noise annoyance surveys may be presented in various ways. This is especially true with respect to the exposure information. Responses may be grouped in noise exposure bins, typically 5 dB wide, or a cluster of respondents may be characterized by the noise level at a centrally located observation point. The number of paired data points may therefore vary substantially from one survey to another.

Some researchers prefer to analyze each survey individually. A regression function is fitted to the survey data, and this function is used to calculate the response at specific exposure levels. A second order polynomial regression function is then calculated to approximate the combined results of different surveys.

A second order polynomial function typically has a minimum (or maximum) value within the relevant exposure range. This contradicts the assumption that the annoyance response increases monotonically with increasing noise exposure levels. Some researchers prefer other functions, (typically, exponential or logistic), or adjust the low end of the calculated exposure–response curve to support a logical explanation of the results. When developing the reference curve used by the EU, for example, Miedema and Vos [52] forced the regression curve through zero at an exposure level *L*_ct_ = 42 dB, in agreement with their observations. A logistic function assumes that the annoyance response asymptotically approaches the end points 0% and 100% highly annoyed.

### 3.2. Community Tolerance Level (CTL) Analysis

This analysis method was introduced by Fidell et al. [53] and Schomer et al. [54] and is described in the international standard ISO 1996-1 [55]. The shape of the exposure–response function is fixed in CTL analysis, rather than derived from a curve fitting analysis. Its shape is based on the observation that the rate of change of annoyance with DNL closely resembles the rate of change of loudness with sound level. The CTL approach accounts for half again as much variance in the relationship between noise exposure and annoyance prevalence rates in communities as conventional analyses [53].

A CTL value, *L*_ct_, for a survey is a predicted value of a noise level, DNL or DENL, at which half of the survey respondents report high annoyance with the noise exposure. Large values of CTL indicate a high tolerance for noise exposure in a community, while small values of CTL indicate a low tolerance for noise exposure. Differences among survey findings may be characterized in decibel-denominated units via simple comparisons of CTL values across communities. The CTL method is therefore well suited to temporal trend analyses. Changes in the CTL value show directly how the prevalence of annoyance varies over time.

The CTL value, *L*_ct_, is a single number that characterize a complete exposure–response curve:HA = 100*exp(−1/((10^((*L*_dn_ − *L*_ct_ + 5.3)/10))^0.3)))(1)

A CTL value is not in itself a noise metric; it is simply a DNL value at which (by convention) half of a community is highly annoyed by noise exposure (and the other half is not.)

Some researchers question whether a common function of the sort assumed by CTL analysis, can be found at all [56]. The “goodness of fit”, i.e., how well a function fits a set of data points, is characterized by the coefficient of determination, r^2^. The two right columns in Table 1 show r^2^ values for the CTL function and a second order polynomial regression curve respectively. There are no meaningful differences between the two values, indicating that from a statistician’s point of view they both fit “equally well” (Student-*t*: average difference r^2^(CTL) − r^2^(poly) = −0.0364 (SD = 0.1778) yielding t(57) = 1.5446 and *p* = 0.128).

In conventional regression analysis, the shape of the dose–response relationship is an artifact of the analysis technique. Logistic regression, for example, yields a dose–response relationship defined by two free parameters (slope and intercept) that are determined by a best fit criterion to a set of data points. CTL analysis yields a dose–response relationship with an assumed pre-defined shape. CTL analysis is therefore more parsimonious than regression analysis, since the single free parameter of CTL analysis is the position of the assumed relationship on the exposure axis.

## 4. Analyses and Discussion

### 4.1. Current EU Reference Curve

The exposure–response function (ERF) that serves as a standard reference in the EU was derived from the results of 26 surveys on road traffic noise [52]. All of these surveys are among those included in Table 1 (marked with M in the EU column).

Figure 1 shows average exposure–response curves for these 26 surveys calculated by standard regression analysis and the CTL method. The figure also shows the exposure–response curve developed by Miedema and Vos [52] currently used by the EU as a standard reference (solid line).

The regular second order polynomial regression function (dashed line) exhibits a typical minimum (at *L*_dn_ = 45.7 dB in Figure 1) and overestimates the annoyance at low exposure levels compared with the reference curve. However, at the high end (at *L*_dn_ > 60 dB) this ERF indicates a lower prevalence of annoyance. In their analysis, Miedema and Vos, explain the adjustments they made at the low end of their ERF. The ERF that is based on CTL analysis underestimates the annoyance for exposure levels around 50–70 dB DNL relative to the EU reference curve. Both the regression curve and the CTL-based curve fall mainly within the 95% confidence interval for the Miedema and Vos curve for values of *L*_dn_ < 75 dB. The 95% confidence interval is about ± 2.5% HA wide at *L*_dn_ = 50 dB and about ± 5% wide at *L*_dn_ = 75 dB [57]. However, the CTL-based curve represents a better fit to the reference curve since its deviation (squared error) is about one-third of that for the regression curve (for the interval indicated above).

### 4.2. Results from 61 Surveys 1969–2014

Figure 2 shows average exposure–response curves for the 61 surveys in Table 1 calculated by both standard regression analysis and by the CTL method analysis. The figure also shows the exposure–response curve developed by Miedema and Vos [52] (solid line). There are no meaningful differences between the ERFs based on the full dataset, 61 surveys, compared with the 26 surveys used for the reference curve (see Figure 1).

### 4.3. Results from Recent Surveys

Some researchers claim that people today are more annoyed by environmental noise than they were, say 25–50 years ago [2,58]. Naturally, as environmental noise levels increase and the population that is exposed to high levels of environmental noise increases, the total number of highly annoyed residents also increases. However, under equivalent exposure conditions, are people more annoyed today than they were in prior years? Is there a need to update the exposure–response functions as WHO has recommended? [59].

Table 1 contains 18 surveys conducted after 2000. The results of these surveys were analyzed separately in the same manner as the previous two datasets. The results are shown in Figure 3.

Using actual observation data when calculating an average regression function may introduce a bias if the number of data points varies among the different studies. In their analysis for WHO, Guski et al. [2] calculated a regression function for each survey individually, and then used these functions to calculate the response at fixed exposure levels. The results from a similar analysis of the 18 post-2000 surveys are shown in Figure 4. The two curves in this figure, the second order regression function and the CTL function display the same goodness of fit to the calculated data points, r^2^ = 0.53.

### 4.4. Annoyance Across a 45-Year Time Span

A CTL value is a single number characterization of the results of a noise annoyance survey. This quantity is therefore well suited for studying temporal trends in the annoyance response. As can be seen from Table 1, CTL values for the 61 surveys in the current data set vary considerably over more than two orders of magnitude, from about 69 dB to 92 dB. The average value is *L*_ct_ 78.4 dB. The range for the corresponding exposure–response curves is shown in Figure 5. The ERF for any one of the 61 surveys lies somewhere between the two solid lines. The average ERF is shown by the dashed red line.

In Figure 6 the CTL value for each survey has been plotted as a function of study year together with two regression lines, a linear and a second order polynomial.

An alternative way to analyze a possible temporal trend is to calculate the exposure level that corresponds to a particular annoyance prevalence rate, for instance 10% highly annoyed. This limit, 10% HA, is suggested by the World Health Organization to prevent adverse health effects [59]. A second order polynomial regression function is fitted to the observation data for each individual survey, and then this function is used to calculate the noise exposure level at which 10% of the exposed population consider itself highly annoyed. The results are shown in Figure 7. The noise levels corresponding to 10% HA has been plotted as a function of study year.

## 5. Conclusions

It has long been known that the noise level per se only accounts for about one-third of the variance in the annoyance response, and that the rest is controlled in a large part by non-acoustic factors [56]. Comparison of the results from different surveys is therefore highly dependent on the type and possible dominance of non-acoustic factors. Nevertheless, ever since Schultz published his original paper on Synthesis of Social Surveys [3], researchers have tried to derive average exposure–response functions. Such functions seem necessary to some for regulatory purposes.

Several attempts have been made to compare prevalence of annoyance rates across continents and different cultures. No dramatic differences have been reported [60]. The database used in the present study is comprised of 61 surveys, 41 of which have been conducted in Europe. The European sub-set has been analyzed separately for comparison.

Community noise regulations in many EU countries typically define the level at which ten percent of the population is highly annoyed, as the limit for acceptable exposure. According to the ERF developed by Miedema and Vos [52], 10% HA corresponds to *L*_dn_ = 58.4 dB, while at *L*_dn_ = 65 dB the prevalence of highly annoyed is 19.4 percent. Table 2 compares of the results of the analyses above.

The ERFs that are based on CTL analysis predicts 10% HA at noise exposure levels about 3 dB greater than the levels predicted by conventional polynomial regression. This is within the predicted confidence interval for the Miedema and Vos curve [57]. The main reason for this difference is that the CTL method assumes an ERF that asymptotically approaches the endpoints of the annoyance scale.

A conventional regression analysis of the 26-survey dataset used by Miedema and Vos yields a slightly different result from that of the original EU reference curve. This is because Miedema and Vos made adjustments to the low end of the ERF curve to force their curve to zero at *L*_dn_ = 42 dB.

The results based on the complete 61-survey data set are very similar to what can be found from the 26-survey data set used by Miedema and Vos. No meaningful difference can be found if the selection of surveys is limited to Europe.

The average results for the 18 post-2000 surveys are not significantly different from previous surveys. The values in Table 2 have been calculated from both the complete data set (see Figure 3) and from regression curves calculated from individual regression functions (see Figure 4). The results actually indicate a small but meaningless decrease in the annoyance response. We therefore conclude that annoyance with traffic noise exposure has been stable within error of measurement and analysis over the past 45 years.

Researchers who rely on statistical methods without any concern for the implications of their findings often fit a regression function to their data to predict possible trends. The second order polynomial function in Figure 6 seems to indicate that the average CTL value has been decreasing from 1970 to about 1990, i.e., people were increasingly annoyed by road traffic noise during this period. However, from 1990 onward, the CTL value increases, indicating that people’s annoyance reaction decreased during the latter period. A similar trend is indicated by Figure 7. The exposure level for 10% highly annoyed decreased from 1970 to about 1990, before again increasing. This seems to indicate that people are becoming more tolerant to noise after a minimum tolerance around 1990.

The r^2^-values for these regression functions are low, (less than 0.15), and no plausible explanation exists for a change in the annoyance response as indicated by the polynomial regression functions. A linear function fitted to the same datasets is nearly horizontal, indicating that the annoyance response has been stable across this 45-year period.

The World Health Organization has recently recommended lowering of the current limit for road traffic noise to 53 dB DNL [59] for purposes of preventing adverse health effects. This limit is about 5–7 dB more stringent than the limits documented above. Only 8 of 61 surveys show a limit for 10%HA of *L*_dn_ = 53 dB or less, and a simple arithmetic average of the results in Figure 7 yields an average limit for 10% HA at *L*_dn_ = 60 dB.

The WHO recommendation is supported only by an analysis by Guski et al. [2] that is an artifact of the selection of surveys included in the analysis [61]. Guski et al. included six surveys from the HYENA study that addressed a limited age range of respondents, 45–70 years [58]. This age group is known to be exceptionally sensitive to noise [44,62,63]. In addition, the analyses of Guski et al. included five surveys from the Austrian Alpine region [64]. These surveys use a different definition for highly annoyed, defining a HA cut-off 60%–64% of the annoyance scale, instead of the commonly used 72%. They can therefore not be readily compared with surveys reported according to standard recommendations [7].

The results from this re-analysis of 61 surveys on road traffic noise conducted over the past 45 years, do not support the decision by WHO to lower the recommended level to avoid adverse health effects. The results indicate that the annoyance due to road traffic noise has been stable across the entire period. This finding is independent of analysis methods.

## Figures and Tables

**Figure 1 ijerph-17-01374-f001:**
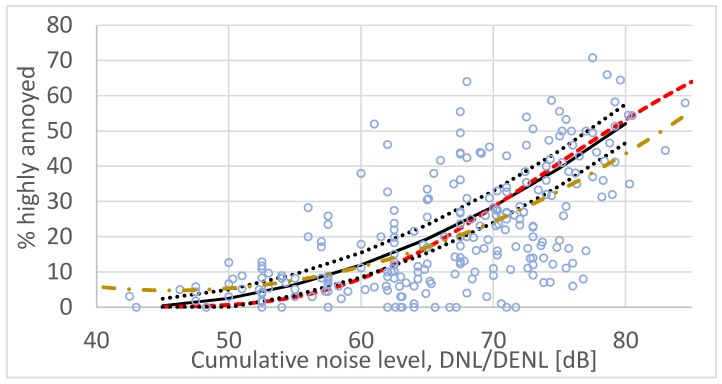
Set of paired exposure/annoyance observations from 26 surveys on road traffic noise (Miedema and Vos data set). Solid line (black) is the exposure–response function (ERF) with corresponding 95% confidence interval (black dotted) developed by Miedema and Vos. Standard second order polynomial regression yields the dash–dot ERF (yellow) while community tolerance level—CTL analysis yields the dashed ERF (red).

**Figure 2 ijerph-17-01374-f002:**
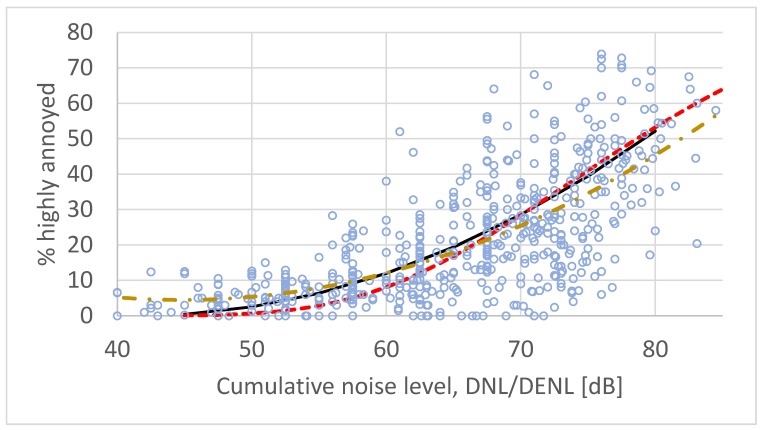
Set of paired exposure/annoyance observations from 61 surveys on road traffic noise (see Table 1). Solid line (black) is the ERF developed by Miedema and Vos. Standard second order polynomial regression yields the dash–dot ERF (yellow) while CTL analysis yields the dashed ERF (red).

**Figure 3 ijerph-17-01374-f003:**
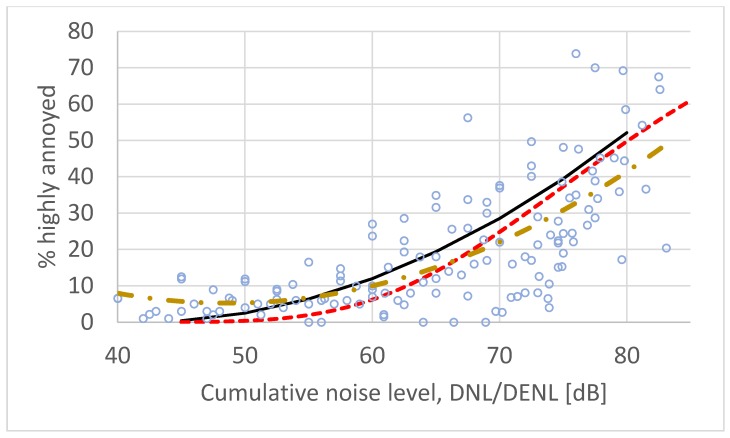
Set of paired data from 18 post-2000 surveys on road traffic noise. Solid line (black) is the ERF developed by Miedema and Vos. Standard second order polynomial regression yields the dashed ERF (yellow) while CTL analysis yields the dotted ERF (red).

**Figure 4 ijerph-17-01374-f004:**
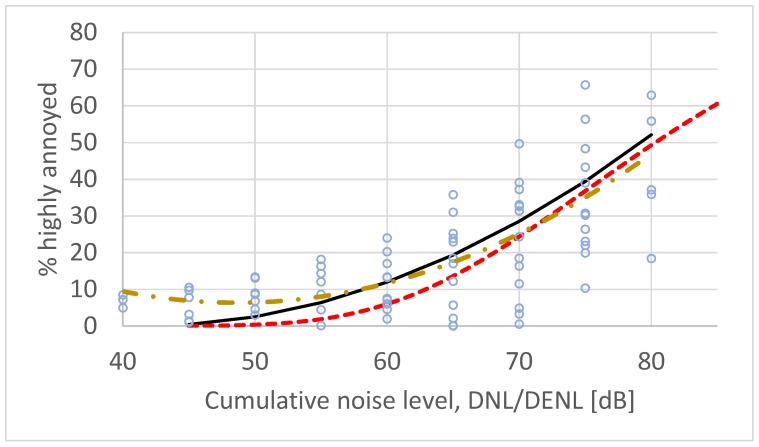
Set of calculated paired exposure/annoyance data from 18 post-2000 surveys on road traffic noise. The datapoints have been calculated from regression functions for each individual survey. Solid line (black) is the ERF developed by Miedema and Vos. Standard second order polynomial regression yields the dash–dot ERF (yellow) while CTL analysis yields the dashed ERF (red).

**Figure 5 ijerph-17-01374-f005:**
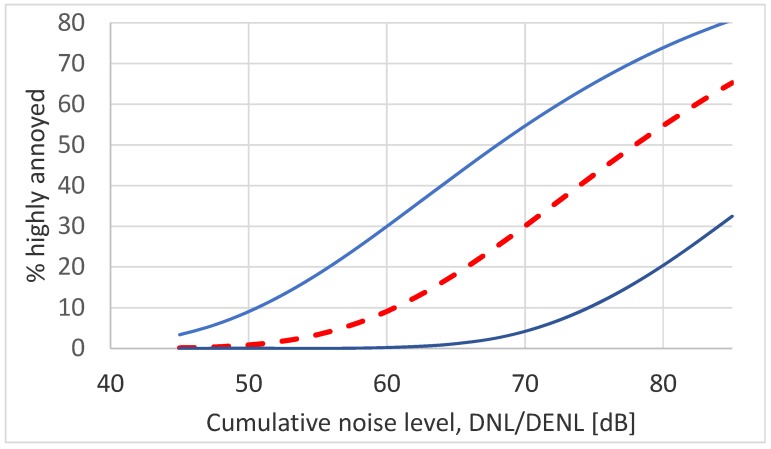
CTL based exposure–response curves for 61 surveys on road traffic noise. The two solid (blue) lines indicate the extremes while the dashed (red) line is the average.

**Figure 6 ijerph-17-01374-f006:**
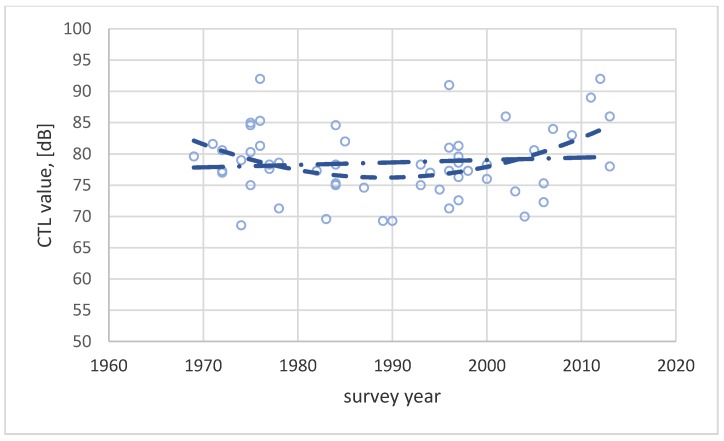
CTL values for 61 surveys on road traffic noise plotted as a function of study year. The two lines show a linear regression function (dash–dotted) and a second order polynomial regression function (dashed).

**Figure 7 ijerph-17-01374-f007:**
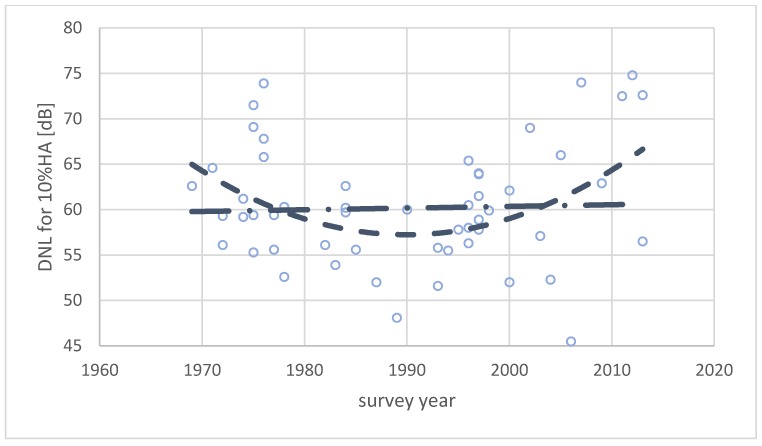
Noise exposure level at which 10% of the respondents are highly annoyed calculated for 61 surveys and plotted as a function of study year. The lines show a linear regression function (dash–dotted) and a second order polynomial regression function (dashed).

**Table 1 ijerph-17-01374-t001:** Annoyance surveys included in the current analysis.

Year	Code	Location	Reference	EU	Resp	CTL	r^2^ CTL	r^2^ poly
1969	FRA041	Paris area	Aubree et al. 1971 [10]		700	79.6	0.95	0.947
1971	SWI053	Basel	Graf et al. 1973 [11]	M	3939	81.6	0.93	0.936
1972	DEN075	Copenhagen	Relster 1975 [12]		960	77.3	0.59	0.711
1972	UKD071	London area	Langdon 1976 [13]	M	2933	77.0	0.65	0.589
1972	UKD072	England	Harland 1977 [14]	M	6017	80.6	0.88	0.872
1974	FRA092	France, 10 areas	Vallet et al. 1978 [15]	M	1000	79.0	0.97	0.990
1974	NET106	Dordrecht	van Dongen 1981 [16]	M	383	68.6	0.96	0.961
1975	BEL122	Antwerp	Myncke et al. 1979 [17]	M	1319	85	0.44	0.151
1975	NET258	Amsterdam	Bitter et al. 1982 [18]	M	622	80.3	0.19	0.785
1975	CAN120	Ontario, 4 areas	Bradley et al. 1977 [19]	M	1150	75.0	0.54	0.053
1975	CAN121	Toronto	Hall et al. 1977 [20]	M	1786	84.6	0.67	0.149
1975	AUL227	Australia	Brown 1978 [21]		818	81.0	0.16	0.20
1976	BEL137	Brussels	Myncke et al.1979 [17]	M	495	81.3	0.67	0.757
1976	SWE142	Sweden, 3 cities	Åhrlin et al. 1979 [22]	M	1377	85.3	0.50	0.796
1976	SWE165	Gothenburg	Åhrlin et al. 1979 [22]	M	464	92.0	0.45	0.754
1977	GER192	Germany, 14 ar.	Knall et al. 1983 [23]	M	1651	77.6	0.77	0.611
1977	UKD157	London, 6 areas	Atkins R&D 1979 [24]	M	1363	78.3	0.90	0.895
1978	CAN168	Canada, 4 areas	Birnie et al. 1980 [25]	M	965	78.6	0.75	0.917
1978	SWI173	Zurich	Nemecek et al. 1978 [26]	M	1607	71.3	0.71	0.750
1982	UKD242	UK, 5 areas	Atkins R&D 1983 [27]	M	2097	77.3	0.59	0.661
1983	NET276	Netherlands, 3ar	Miedema et al. 1985 [28]	M	798	69.6	0.36	0.828
1984	FRA239	France, 4 areas	Diamond et al. 1986 [29]	M	1032	84.6	0.10	0.451
1984	NET240	Schiphol	Diamond et al. 1986 [29]	M	581	75.3	0.64	0.842
1984	NET362	Arnhem, 3 areas	Ericz et al. 1986 [30]	M	1322	75.0	0.96	0.982
1984	UKD238	Glasgow	Atkinson et al. 1985 [31]	M	608	78.3	0.88	0.913
1985	GER372	Germany 2 cities	Paulsen et al. 1986 [32]	M	564	82.0	0.54	0.220
1987	GER373	Ratingen	Kastka et al. 1996 [33]	M	516	74.6	0.99	0.999
1989	AUS329	Alpine area	Lercher et al. 1993 [34]		1989	69.3	0.92	0.908
1990	ITL350	Italy, Modena	Bertoni et al. 1994 [35]		908	69.3	0.99	0.959
1993	NET361	Netherlands	de Jong et al. 1994 [36]	M	4038	75.0	0.85	0.874
1993	FRA364	France, 18 ar	Vallet et al. 1996 [37]	M	895	78.3	0.65	0.858
1994		Kumamoto	SASDA JPN003RT [38]		387	77.0	0.93	0.973
1995	TRK367	Istanbul	Kurra et al. 1995 [39]		154	74.3	0.99	0.999
1996	SWE368	Gothenburg, det	Sato et al. 1998 [40]		436	71.3	0.96	0.973
1996	SWE368	Gothenburg, ap	Sato et al. 1998 [40]		706	81.0	0.89	0.923
1996	JPN369	Kumamoto	Sato et al. 1998 [40]		378	77.3	0.91	0.997
1996	JPN369	Kumamoto	Sato et al. 1998 [40]		458	91.0	0.99	0.999
1997	BAN97	Bangkok	Sato et al. 1998 [40]		183	72.6	0.98	0.977
1997	FRA394	Besancon	Houot 1999 [41]		1910	78.6	0.91	0.920
1997		France, 61 areas	Cremezi et al.2001 [42]		673	81.3	0.95	0.944
1997	JPN382	Sapporo, det	Sato et al. 1998 [40]		411	79.6	0.95	0.999
1997	JPN382	Sapporo, apt	Sato et al. 1998 [40]		369	76.3	0.99	0.999
1998		Kanagawa	SASDA JPN009RT [38]		353	77.3	0.95	0.962
2000		Japan, 8 cities	SASDA JPN011RT [38]		1056	76.0	0.92	0.939
2000	SWE526	Sweden, 2 cities	Öhrström et al.2004 [43]		956	78.3	0.88	0.863
2002	DEN529	Copenhagen	Bjørner 2004 [44]		1149	86.0	0.99	0.996
2003		Tomakomai	SASDA JPN016RT [38]		1056	74.0	0.93	0.980
2003		Copenhagen	Vejdirektoratet 2013 [45]		2870	76.2	*)	
2004		Kanagawa	SASDA JPN021RT [38]		1576	70.0	0.93	0.906
2005		Hanoi	Shimoyama et al. 2014 [46]		1503	80.6	0.39	0.294
2006		Alp. motorway	Lercher et al. 2008 [47]		3630	75.3	0.58	0.804
2006		Alp. Main road	Lercher et al. 2008 [47]		3630	72.3	0.98	0.979
2007		Ho Chi Minh	Shimoyama et al. 2014 [46]		1471	84.0	0.13	0.377
2009		Hong Kong	Brown et al. 2015 [48]		10077	83.0	0.92	0.912
2011		Da Nang	Shimoyama et al. 2014 [46]		492	89.0	0.97	0.984
2012		Hue	Shimoyama et al. 2014 [46]		688	92.0	0.72	0.727
2012		Stockholm	Länsstyrelsen, 2016 [49]		3086	76.6	0.83	0.961
2013		Thai Nguyen	Shimoyama et al. 2014 [46]		813	86.0	0.63	0.721
2013		Swiss, multi-site	Brink et al. 2015 [50]		2386	78.0	0.95	0.983
2014	Den	Urban roads	Vejdirektoratet, 2016 [51]		3315	79.6	*)	
2014	Den	Motorways	Vejdirektoratet, 2016 [51]		3446	67.6	*)	

*) No single data points, only exposure–response function (ERF) available.

**Table 2 ijerph-17-01374-t002:** Comparison of the analysis results from different selections of surveys.

Data Set	Poly 10% HA	CTL 10% HA	Poly 65 dB DNL	CTL 65 dB DNL
26 surveys, Miedema and Vos	58.4 dB DNL	61.3 dB DNL	17.0% HA	16.6% HA
61 surveys, 1969–2013	57.8 dB DNL	61.2 dB DNL	17.8% HA	16.9% HA
41 European, 1969–2013	57.6 dB DNL	60.7 dB DNL	18.3% HA	18.2% HA
18 surveys, 2000–2013	59.8 dB DNL	62.9 dB DNL	15.3% HA	13.6% HA
18 surveys, individual reg.	58.0 dB DNL	62.8 dB DNL	17.5% HA	14.0% HA

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
