# Peer review of "On the Temporal Stability of People’s Annoyance with Road Traffic Noise"

_ijerph, 2020, doi:10.3390/ijerph17041374_

Round 1

Reviewer 1 Report

Thank you for providing the opportunity to review this paper.

This study analyzes the temporal stability of people’s annoyance with road traffic noise. It is an interesting read and I do not have any major concerns. However, I would suggest the following:

The abstract is concise and good but Introduction should provide more details particularly summarizing previous studies on noise, attention diversion, and health implications.

Many acronyms are not defined.

Is there really a reason to add ‘Code’ in Table 1? Does it add any value?

Some of the information provided in section 3.1 should be moved into the Introduction section such as Line 59 to 66.

I do not have any comments or suggestions on Discussion and Conclusion.

Author Response

Thank you for reviewing my paper and providing comments that would improve it. 

The introduction has been expanded, as suggested, by moving some parts from section 3.1 dealing with the definitions of annoyance and annoyance scales.

All acronyms have been fully explained the first time they appear in the text.

The "code" in table 1 provides a unique ID for the survey and is very useful for survey identification. In addition a separate bibliography explaining the origin of all survey data has been added. This information is useful for readers that want more detailed information about the individual surveys, but it would expand the paper too much if this information should be included in the actual text.

Reviewer 2 Report

This manuscript is interesting and the argument and analysis are sound. Please consider to make revisions according to the following comments.

1 Please add a detailed introduction on the contents of all these sureys or data samples. Especially the survey methodology for each detailed case, such like the tolerance for traffic noise for passengers during traveling or for tourists when taking a rest at the roadside, is required for readers' better understanding. 

2 Please add the legends description in all figures. 

3 Do the multiple sources of traffic noice need to be seperated or being set various recommendations?

4 All the abbreviation words should be given their original form.

Author Response

Thank you for providing comments that would improve my paper.

A detailed description of all surveys that have been analyzed, would require a large expansion of the paper (many pages), and it is not necessary for following the main conclusions. Instead I have included detailed bibliographic information about reports/articles/papers where the survey information can be found. The Wyle catalogue with the survey ID-code is a good start.

All figure captions have been checked that they have the legends included and described.

All acronyms have been described at their first appearance in the text.